# Ceruloplasmin Reduces the Lactoferrin/Oleic Acid Antitumor Complex-Mediated Release of Heme-Containing Proteins from Blood Cells

**DOI:** 10.3390/ijms242316711

**Published:** 2023-11-24

**Authors:** Anna Yu. Elizarova, Alexey V. Sokolov, Vadim B. Vasilyev

**Affiliations:** Institute of Experimental Medicine, 197376 Saint-Petersburg, Russia; anechka_v@list.ru (A.Y.E.); vadim@biokemis.ru (V.B.V.)

**Keywords:** lactoferrin, ceruloplasmin, oleic acid, protein–lipid complex, antitumor complex

## Abstract

Our previous study showed that not only bovine lactoferrin (LF), the protein of milk and neutrophils, but also the human species forms complexes with oleic acid (OA) that inhibit tumor growth. Repeated injections of human LF in complex with OA (LF/8OA) to hepatoma-carrying mice decelerated tumor growth and increased animals’ longevity. However, whether the effect of the LF/8OA complex is directed exclusively against malignant cells was not studied. Hence, its effect on normal blood cells was assayed, along with its possible modulation of ceruloplasmin (CP), the preferred partner of LF among plasma proteins. The complex LF/8OA (6 μM) caused hemolysis, unlike LF alone or BSA/8OA (250 μM). The activation of neutrophils with exocytosis of myeloperoxidase (MPO), a potent oxidant, was induced by 1 μM LF/8OA, whereas BSA/8OA had a similar effect at a concentration increased by an order. The egress of heme-containing proteins, i.e., MPO and hemoglobin, from blood cells affected by LF/8OA was followed by a pronounced oxidative/halogenating stress. CP, which is the natural inhibitor of MPO, added at a concentration of 2 mol per 1 mol of LF/8OA abrogated its cytotoxic effect. It seems likely that CP can be used effectively in regulating the LF/8OA complex’s antitumor activity.

## 1. Introduction

Most of the antitumor drugs currently used to treat cancer display toxic effects on both growing tumors and proliferating normal cells, which occur partly due to free radical production [1]. The fortuitous discovery, in 1995, of a complex formed by alpha-lactalbumin (α-LA) and oleic acid (OA), later called HAMLET (Human Alpha-lactalbumin Made LEthal to Tumor cells), prompted many researchers to study protein–lipid complexes as promising antitumor compounds [2]. Numerous in vitro models [3,4], animal models of cancer [5,6,7], and clinical trials [8,9] have demonstrated the antitumor activity of HAMLET. A synthetic version of HAMLET, produced by Swedish HAMLET Pharma and called Alpha1H, is currently being used in a clinical study [10]. Since the first discovery of the complex formed in breast milk by α-LA and OA, about ten proteins capable of forming HAMLET-like complexes have been discovered, all of which efficiently destruct tumor cells [11,12,13,14,15]. Lactoferrin (LF) is one such protein [16,17]. It is a non-heme iron-binding glycoprotein and a member of the transferrin family with a molecular mass of 76–80 kDa [18]. It is predominantly present in milk (1–4 mg/mL) and colostrum (7 mg/mL) but also found in exocrine secretions [19,20]. LF interacts with cancer cells, showing no marked side effects, such as allergic or an autoimmune responses. It can cross the blood–brain and blood–tissue barriers, and its antitumor effect is aimed at a broad spectrum of molecular targets that control the proliferation, survival, migration, invasion, and metastasis of tumor cells [21,22,23,24].

We have shown previously that mixing LF and OA in an ethanol-containing solution resulted in the formation of a complex that exhibits in vitro and in vivo antitumor activity towards hepatoma 22a cells [25]. Additionally, the capacity of the LF/8OA complex to cause the lysis of erythrocytes has been demonstrated [25]. Since mice receiving a daily dose of 5 mg of LF/8OA did not demonstrate drug toxicity after 22 days, we have further investigated how the complex reduces cytotoxicity. LF is known to form a complex with the copper-containing plasma protein ceruloplasmin (CP). We hypothesized that OA will not interfere in the interaction between LF and CP. CP has a molecular mass of 132 kDa and is one of the acute-phase reactants in inflammation [26,27]. It protects erythrocytes from copper-induced lysis and is used in the treatment of various types of anemia [28]. Thus, we studied the effect of CP on the survival of neutrophils, as its capacity to mitigate their activation was shown in our previous study [29].

It remained unclear whether any protein appearing in plasma, e.g., serum albumin, is equally toxic upon binding fatty acids in comparison with LF, as administering injections of the LF/8OA complex to tumor-carrying mice and to control animals had no lethal effect, suggesting the existence of a protective factor in murine blood.

This study compared the impact of LF and BSA, carrying OA, on the lysis of human and murine erythrocytes and on the activation of human neutrophils, which results in the egress of myeloperoxidase and, consequently, oxidative (halogenating) stress. In view of the high affinity of CP towards LF/8OA, we studied its capacity to modify, in a dose-dependent manner, the effect of LF/8OA on erythrocytes and neutrophils in vitro.

## 2. Results and Discussion

### 2.1. Interaction of Ceruloplasmin with Lactoferrin and with the Lactoferrin/Oleic Acid Complex

The effect of LF/8OA on blood cells might be neutralized by plasma proteins; hence, we analyzed its interaction with CP, which is the principal partner of LF. LF and LF/8OA were added either to pure human and murine CP or to samples of the respective sera, and mixtures obtained were subjected to PAGE without detergents. Staining gel with *o*-DA allowed us to detect CP and compare its ability to form a complex with LF and/or LF/8OA (Figure 1).

Both LF and LF/8OA equally shifted the *o*-DA-colored band of CP. The human and murine CP did not differ in their interaction with LF/8OA, and furthermore, a fluctuation in mobility, typical of CP interacting with LF, was observed upon adding human or murine serum instead of purified protein. Therefore, CP is a feasible factor for modulating the effects of LF/8OA on normal and possibly malignant cells. The interaction of human CP with LF in comparison with LF/8OA was studied using SPR (see below). This approach was not applicable to murine CP due to its instability under conditions of immobilization on biosensors and their regeneration.

### 2.2. Ceruloplasmin Interacts with Lactoferrin and with the Lactoferrin/Oleic Acid Complex: Kinetics of Interaction

Sensograms characterizing the interaction between LF or LF/8OA (3.125–12.5 μM) and human CP immobilized on a CM5 chip are shown in Figure 2a,b. The dose-dependent increase in the interaction of both LF and LF/8OA with CP was virtually the same due to the close kinetic values of the complex formation. This concerns the dissociation rate constant (k_d_), association rate constant (k_a_), and the maximum binding capacity (R_max_) of the analyte surface.

All values obtained are similar for LF and LF/8OA (see Table 1). It can be concluded that adding OA to LF has no statistically significant effect on its capacity to form a complex with CP. The values obtained for K_D_, characterizing the affinity of CP to LF and to its complex with OA, showed minor non-significant differences—0.367 μM (CP–LF) and 0.407 μM (CP–LF/8OA).

### 2.3. Comparing the Effect of Complexes of Lactoferrin or Albumin with Oleic Acid on the Lysis of Erythrocytes

The effect of the preparations on the lysis of erythrocytes was assayed in vitro when LF and LF/8OA were added to a suspension of red blood cells (1 × 10^4^ cells per well). Figure 3 presents a diagram showing that LF alone has no hemolytic activity in the range of concentrations applied (2–128 μM). In contrast, 8 μM LF/8OA caused the lysis of 40% of the erythrocytes (*p* < 0.01), after 1 h of incubation. At concentrations ranging from 12 μM to 128 μM LF/8OA, cell lysis was induced by >90% compared with the suspension of intact erythrocytes (*p* < 0.01).

Cells in culture are devoid of protective and adaptive mechanisms that can be activated in the body. The experiments with isolated erythrocytes provided evidence that CP, an acute phase reactant and antioxidant, protects erythrocytes from lysis induced by LF/8OA. Figure 4 shows that the incubation of CP and LF/8OA with a suspension of erythrocytes (1 h, 37 °C) mitigated the hemolytic effect (*p* < 0.05). Adding 0.5 mol CP to 1.0 mol LF reduced the number of lysed cells by 24.3 ± 1.4%. Finally, a 1:1 molar proportion of CP and LF/8OA resulted in virtually the full inhibition of hemolysis (97.4 ± 1.3%).

To explore the possible effect of water-insoluble OA on cells, its complex with BSA (the principal carrier of unesterified fatty acids in plasma) was tested under the same conditions. This complex has low hemolytic activity in comparison with LF/8OA (Figure 5). At the highest concentration of 256 μM, BSA/8OA hemolytic activity was 14 times lower than LF/8OA, which caused the lysis of 100% of the erythrocytes.

### 2.4. The Activation of Neutrophils in the Presence of the LF/OA Complex

The degranulation of neutrophils in vitro was determined by measuring the amounts of myeloperoxidase (MPO) released from cells. MPO was measured using ELISA in the supernatant of neutrophils’ suspension (3 × 10^5^ cells per well) after 1 h of incubation with LF or LF/8OA in the presence of glucose. Figure 6 shows that treatment with LF (−36 μM) had no significant activating effect on the neutrophils compared with the intact cells (*p* ˃ 0.05), as no release of MPO was observed. Meanwhile, adding LF/8OA in concentrations ranging from 1 to 36 μM resulted in a significant dose-dependent increase in the release of MPO from neutrophils in comparison with both the intact (*p* < 0.01) and LF-treated cells (*p* < 0.01).

In order to facilitate a comparison with the results derived from studying the protective effect of CP on the red blood cells lysed by LF/8OA, we examined its capacity to prevent the toxic effect of the complex on human neutrophils. The incubation of neutrophils (3 × 10^5^ cells per well) with 64 μM LF/8OA in the presence of CP resulted in a significantly decreased release of MPO compared to the incubation of cells with 64 μM LF/8OA without CP (*p* < 0.05). Importantly, the molar relation CP:LF/8OA less than 1:2 decreased the release of MPO (Figure 7).

The complex formed by BSA and OA had a less pronounced effect on the degranulation of neutrophils compared with LF/8OA (Table 2). Indeed, testing LF/8OA or BSA/8OA in concentrations ranging from 1 to 18 μM showed that the amount of MPO released after treatment with LF/8OA was three times higher when compared with that detected upon incubation with BSA/8OA (Figure 8). This difference became smaller with increasing concentrations above 18 μM (Figure 8). Therefore, LF/8OA tested in vitro showed a dose-dependent cytotoxicity towards neutrophils that was several-fold higher than the activity demonstrated by BSA/8OA under similar conditions.

The results obtained suggest that the anti-tumor mechanisms of LF/8OA described in our previous paper [25] include MPO exocytosis from activated neutrophils followed by pronounced oxidative/halogenating stress [30,31]. Reciprocally, a growing tumor induces CP synthesis [32], and its MPO-inhibiting effect seems to mitigate the toxicity of LF/8OA directed against the malignant cells. The absence of the acute toxicity of LF/8OA injected into mice in amounts as high as 4 g/kg is likely to be also mediated by CP, which cannot be attributed to MPO inhibition and needs to be studied further [25]. Similarly, the decrease in the cytotoxic effect of LF/8OA on neutrophils in the presence of CP cannot be explained by its capacity to inhibit the halogenating activity of MPO [33,34].

It was shown recently that tumor-associated neutrophils produce CP, enabling their survival [35]. The priming of neutrophils is caused by CP in localized aggressive periodontitis [36]. In the case of malignant growth plasma, the concentration of CP increases from 3 to 10 μM, which is sufficient for inhibiting the cytotoxicity of the LF/8OA complex, as shown by the results obtained in this study. In clinical practice, a concentration of LF/8OA complex in tissues is needed to achieve an antitumor effect that can be provided pharmacologically (e.g., via subcutaneous injections).

Our study demonstrated that CP reduced the egress of MPO from activated neutrophils, but the details of this effect deserve a detailed investigation. The uncoupling of MPO and CP can be caused by autoantibodies against MPO, which results in the development of systemic vasculitis, a complication of autoimmune pathology [37]. The latter features noticeable changes in the MPO glycosylation profile [38]. Meanwhile, CP was shown to affect MPO activity via reshaping its glycosylation; e.g., hyper-truncated Asn355-glycans augment the ceruloplasmin-mediated MPO inhibition [39].

It can be suggested that modulating the CP–LF interaction will alter the antitumor features of LF/8OA complex. We have shown that the CP–LF complex is dissociated by heparin, DNA, and lipopolysaccharides, along with the RRRR peptide [26] mimicking the N-terminal polyarginine cluster ^2^RRRR^5^ that links LF with all the anions mentioned [40,41,42]. Therefore, introducing LF with a truncated N-terminus may disrupt its interaction with CP and amplify the cytotoxicity of the LF/8OA complex.

CP is known to provide copper to malignant cells, as it is needed for neovascularization [32,43]. Hence, using chelators for copper deprivation is likely to hamper tumor growth and malignization itself [44,45]. Indeed, antitumor features have been documented for ammonium thiomolybdate, which is associated with neo-angiogenesis [46,47]. Copper depletion seems to disturb the tertiary structure of CP, which is typical for copper’s deprivation caused by an AgCl-saturated diet [48,49]. Therefore, such a protein will not interact with LF. Such a hypothesis requires experimental verification via testing the antitumor features of the LF/8OA complex in combination with copper deprivation.

## 3. Materials and Methods

Recombinant human LF purified from the milk of transgenic goats was a generous gift from our colleagues from the Scientific Practical Center of the National Academy of Sciences of Belarus (Zhodino, Belarus) [50]. Murine monoclonal antibodies (Mab) 1#8 and 2#7 against myeloperoxidase (MPO) were obtained using hybridoma technology, and Mab 2#7 was conjugated with sodium periodate-oxidized horseradish peroxidase, with a subsequent reduction in Schiff bases being carried out using sodium borohydride [51].

The following buffer solutions were used: PBS (0.15 M NaCl, 10 mM sodium-phosphate buffer, pH 7.4) and HBS-P+ (150 mM NaCl, 10 mM Hepes buffer, pH 7.5, 0.05% poly-oxyethylene sorbitan). All solutions were filtered using syringes through sterile membranes of mixed cellulose esters (Syringe Filter, MEC, “Jet Biofil”, Guangzhou, China; pore size 0.22 μM, if not specified).

All protocols of sampling healthy donors’ whole blood were approved by the Local Ethical Committee of the Institute of Experimental Medicine (protocol 1/20 of 27 February 2020).

### 3.1. Isolation and Purification of Human and Murine Ceruloplasmin

To obtain monomeric CP, human blood plasma containing 1 mM EDTA and 0.1 mM phenylmethyl sulfonyl fluoride was subjected to ion exchange chromatography on UNOsphere Q (“BioRad”, Hercules, CA, USA) and to affinity chromatography on neomycin-agarose [52]. Thrombin traces were eliminated on a column with benzamidine-agarose (“Sigma”, St. Louis, MO, USA). Thus, the obtained human CP had A_610_/A_280_ = 0.049 corresponding to 99% purity. A similar protocol was applied to purify 34 mg of CP (A_610_/A_280_ = 0.048) from 320 mL of murine serum. Along with assaying A_610_/A_280_ relation, the homogeneity of CP was confirmed via SDS electrophoresis. In both the human and murine CP preparations, the original buffer was substituted with HBS in three concentration/dilution cycles using Vivaspin 20 centrifuge units with a cut-off 100 kDA (Sartorius, Göttingen, Germany). The resulting solutions contained 500 μM CP (66 mg/mL) and were stored at −80 °C in aliquots of 10–20 μL.

### 3.2. Isolation and Purification of Myeloperoxidase 

MPO was isolated from a human leukocytic foam coat via sequential chromatographic fractionating on heparin-Sepharose, phenyl-Sepharose, and Sephadex G 150 Superfine (“Pharmacia”, Uppsala, Sweden) [53]. The obtained preparation had an A_430_/A_280_ relation of no less than 0.75. Purified MPO (16 μM) was divided into aliquots of 50–200 μL and stored at −20 °C.

### 3.3. Spectrophotometry for Measuring Concentration of Proteins 

The absorption spectra of the preparations were measured using a spectrophotometer SF-2000-02 (“OKB-Spektr”, Sankt-Peterburg, Russia). The concentration of proteins was determined using the following molar extinction coefficients: CP − ε_610_ = 10,000 M^−1^ × cm^−1^ [54]; LF − ε_280_ = 85,700 M^−1^ × cm^−1^ [55]; BSA − ε_280_ = 43,824 M^−1^ × cm^−1^ [56]; MPO − ε_430_ = 178,000 M^−1^ × cm^−1^ [30].

### 3.4. Obtaining Complexes of Proteins with Oleic Acid

A complex of LF with OA (Sigma-Aldrich (Saint Louis, MO, USA)) was obtained via simple mixing, carried out as described previously [25]. A similar protocol was used to obtain a complex of BSA (“Amresco”, Solon, OH, USA) and OA. First, 1 mL of PBS and 0.1 mL of ethanol were carefully layered on 4 mL of 500 μM protein dissolved in PBS. Mixing in a vortex followed, after which 25 μL of 51.2 μM OA in ethanol (14.4 mg/mL) was added to provide the relation of 1 mol OA per 1 mol protein. Every addition was followed by three 30 min energetic mixings at room temperature with 1 min intervals, achieving an OA/protein concentration of 8:1. Excess ethanol and OA was removed via overnight dialysis against PBS at +4 °C and subsequent filtering through a Syringe Filter (Jet Biofil, China) with a pore size 0.45 μm. In the control, LF and BSA were mixed with ethanol without OA; dialysis and filtering were repeated thrice to test whether such manipulations and ethanol itself affected the results. In the control experiments, CP with OA were mixed, but OA did not interact with CP and formed micelles after being added.

### 3.5. Analysis of Stoichiometry in Complexes Formed by LF or BSA with OA

In vitro measurements of OA bound to LF or BSA complexes were measured. Their dialysis and filtration were carried out using the NEFA kit for the colorimetric enzymatic method (“Randox”, Crumlin, UK). The latter is based on the acyl-coenzyme A synthase-driven transformation of non-esterified fatty acids in the presence of ATP and coenzyme A into acetyl coenzyme A, AMP and pyrophosphate, followed by acetyl coenzyme A oxidation catalyzed by specific oxidase, with hydrogen peroxide as the end product. The latter peroxidase-catalyzed reaction involving 4-aminopteridine and 3-methyl-N-ethyl-N-(beta-hydroxyethyl)-aniline resulted in the formation of a purple pigment with a maximum absorption of 550 nm. All measurements were carried out in accordance with the manufacturer’s instructions.

### 3.6. Isolation of Erythrocytes 

Whole blood was sampled from healthy donors into vacutainers (Weihai Hongyu Medical Devices Co., Ltd., Weihai, China) with EDTA as the anticoagulant. Blood was centrifuged at 1000× *g* for 10 min, after which plasma and leukocytic layer were discarded. Washing erythrocytes with PBS thrice allowed for the elimination of most of the remaining plasma. Then, the cells were centrifuged (1000× *g*, 10 min) and immediately used in the experiment.

### 3.7. Analysis of Hemolytic Activity 

A total of 200 μL of LF, BSA, LF/8OA, BSA/8OA (4–256 μM) in PBS and 20 μL of erythrocytes’ suspension in PBS were added to the wells of flat-bottom 96-well plates (“Nuova Aptaca SRL”, Canelli, Italy), 1 × 10^4^ cells per well. The plates were kept for an hour in a thermal shaker PST-60HL-4 (“Biosan”, Riga, Latvia) at 300 r.p.m. and 37 °C. The positive hemolysis control was achieved by substituting the PBS in the wells with pyrogen-free deionized water. This incubation was followed by 5 min of centrifugation of the plates at 1500× *g*, and the supernatant was carefully removed without touching the erythrocytes. Human CP was applied at concentrations of 16–256 μM to evaluate its effect on the hemolysis caused by 128 μM LF/8OA. The same effect of murine CP (1–64 μM) was assayed using 8 μM of LF/OA.

A plate spectrophotometer CLARIOstar (“BMG LABTECH”, Ortenberg, Germany) enabled the evaluation of hemolysis based on the intensity of absorption at A_412_ (the Soret band in hemoglobin). The degree of hemolysis was calculated using the formula (At − Ac)/(Af − Ac) × 100%, where At stands for A_412_ in supernatant, Ac is A_412_ in the supernatant without admixtures (negative control), and Af is A_412_ in the fully lysed erythrocytes (positive control).

### 3.8. Isolation of Neutrophils

Neutrophils were isolated from fresh donor blood from which erythrocytes had been precipitated with dextran T-70 (“Vekton”, Saint-Petersburg, Russia). The leukocyte-enriched plasma was centrifuged in a histopack-1077 solution (“Merck”, Darmstadt, Germany), and trace erythrocytes were eliminated via hypotonic lysis [57]. Precipitated Nph were washed with PBS containing 2 mg/mL D-glucose. The suspension obtained (30–80 × 10^6^ Nph/mL) was stored at 4 °C for no longer than 1–2 h. The percentage of Nph moieties in the suspension was 97–98%, while the percentage of viable cells was no less than 98% (tested using trypan blue). All experiments involving Nph were completed on the day of blood sampling.

### 3.9. Myeloperoxidase Exocytosis

LF, BSA, LF/8OA, abd BSA/8OA (1–36 μM) in PBS containing 2 mg/mL D-glucose were placed in 96-well plates (3 × 10^5^ cells per well). The plates were placed in a thermal shaker, PST-60HL-4 (“Biosan”, Riga, Latvia), for an hour at 300 r.p.m and 37 °C. Then, the plates were centrifuged at 1500× *g* for 5 min, and the supernatant was collected to measure MPO concentration using ELISA with Mabs to MPO. 1#8 Mab was immobilized on a solid phase, and horseradish peroxidase-labeled 2#7 Mab was used to detect MPO [58]. Human CP was used at concentrations of 2–128 μM to evaluate its effect on MPO exocytosis caused by 64 μM LF/8OA.

### 3.10. Studying Protein–Protein Interaction Using Biosensors Based on Surface Plasmon Resonance 

To investigate the interaction strength within the protein–protein complexes formed by CP and LF (or LF/8OA) and to determine the equilibrium constant of a complex formation, we used surface plasmon resonance (SPR). The interaction between molecules was assayed on Biacore X100 (“GE-Healthcare”, Hatfield, UK) with a standard CM5 (carboxymethyl-dextran) chip covered with carboxymethylated dextran. The chip was activated by an equimolar mixture of N-ethyl-N′-dimethyl aminopropyl carbodiimide (EDC) and N-hydroxy succinimide (NHS; 0.2 M), after which CP (20 μg/mL) in 0.5 mM sodium-acetate buffer, pH 5.5, was injected. The buffer flow rate was 5 μL/min for a 12 min immobilization. To block the remaining unbound groups on the chip, 1 M ethanolamine-HCl, pH 8.5, was added. The resulting immobilization was 3.3 ng of CP per 1 mm^2^ of the chip.

Increasing concentrations of the analyte (A), i.e., LF, were injected over a CP-covered chip at a flow rate of HBS-P+ buffer 30 μL/min. Each analysis consisted of four stages: (1) injection of HBS-P+ buffer for 1 min; (2) injection of A (LF or LF/8OA) for 1 min; (3) dissociation for 5 min (washing with HBS-P+ without A); (4) regeneration in 90 s with HBS-P+ containing 1 M NaCl and 0.1 M ethanolamine. The part of the flow cell devoid of immobilized CP was performed as a control for assaying the non-specific binding of A with the chip (˂2% signal in case of the maximum saturation with A). A binding curve obtained for each concentration of A was subtracted from the respective curves of CP binding with A. The equilibrium dissociation constant K_D_ for CP–LF (CP–LF/8OA) was determined by a stable response (RU) on A concentration. The data obtained were consistent with the equation of Langmuir, describing a single-center binding model (1:1):$$\frac{R_{eq}}{R_{max}}=\frac{\left[A\right]}{\left[A\right]+K_{D}}$$
where R_eq_—response value (RU) at equilibrium; R_max_—response (RU) at surface saturation with analyte; K_D_—equilibrium dissociation constant.

### 3.11. Electrophoretic Methods 

Various electrophoretic procedures were used to analyze homogeneity, molecular mass, and electrophoretic mobility in polyacrylamide gel (PAG). Native proteins were subjected to PAGE in 7.5% resolving gel containing Tris–HCl buffer with pH of either 8.9 [59] or 7.5 [60]. The specific oxidase activity of CP was measured by soaking gels in a solution of o-dianisidine (o-DA) (“Sigma”, St. Louis, MO, USA) [61]. PAGE in Tris–HCl buffer with SDS was used to analyze molecular mass and protein purity [62].

### 3.12. Statistical Analysis

The experimental data were processed using Microsoft Excel 2008 and are presented as mean ± standard error of the mean. Statistical significance was evaluated using the Mann–Whitney U-test for non-parametric samplings. In the case of normal distribution, Student’s test was used. Significance at *p* ≤ 0.05 was considered positive.

## 4. Conclusions

It seems likely that the interaction of CP with LF can be a reliable target for the regulation of the LF/8OA complex’s antitumor activity. Probably, abrogating the interaction of CP with LF within that complex might amplify the anti-tumor effect of LF/8OA. On the other hand, it should be clarified whether CP in the absence of its interaction with LF retains the capacity to protect non-malignant cells against LF/8OA. In view of CP contribution in loading LF with iron [63] and the incapacity of iron-saturated LF to stabilize hypoxia-inducible factor 1-alpha while inducing ferroptosis in tumor cells [64], we suggest that Fe-LF/8OA will demonstrate more pronounced antitumor features, and we plan to test this hypothesis in future studies.

## Figures and Tables

**Figure 1 ijms-24-16711-f001:**
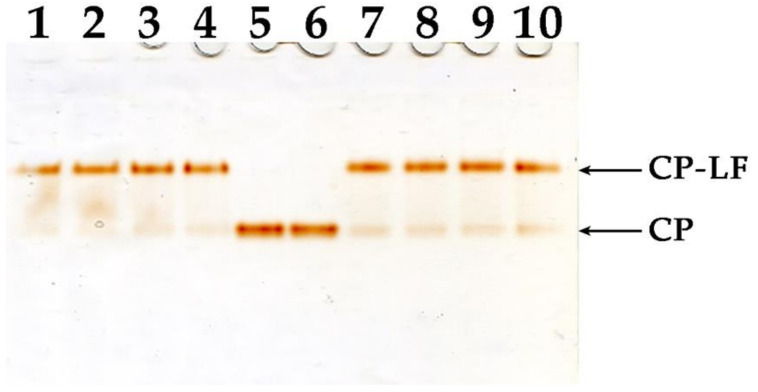
Disc electrophoresis of LF-CP and LF/8OA-CP complexes in PAG without detergents. Staining with o-dianisidine. 1—CP + LF (8 μg); 2—CP + LF (4 μg); 3—CP + LF (2 μg); 4—CP + LF (1 μg); 5, 6—CP (1 μg/lane); 7—CP + LF/8OA (1 μg); 8—CP + LF/8OA (2 μg); 9—CP + LF/8OA (4 μg); 10—CP + LF/8OA (8 μg).

**Figure 2 ijms-24-16711-f002:**
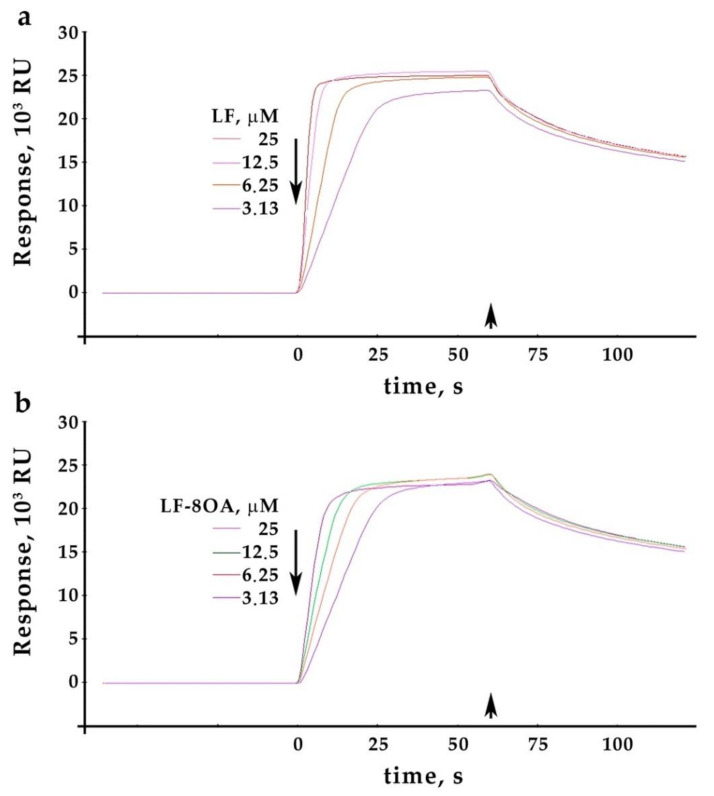
Interaction of CP with LF and its complexes with OA obtained using the surface plasmon resonance technique on Biacore X-100: (**a**,**b**)—sensograms of interaction of LF and LF/8OA with immobilized CP (3.125—25.0 nM). By abscissa—time, sec.; by ordinate—signal of biosensor, RU (resonance units). Long arrow marks the analyte adding; short arrow marks adding analyte-free buffer.

**Figure 3 ijms-24-16711-f003:**
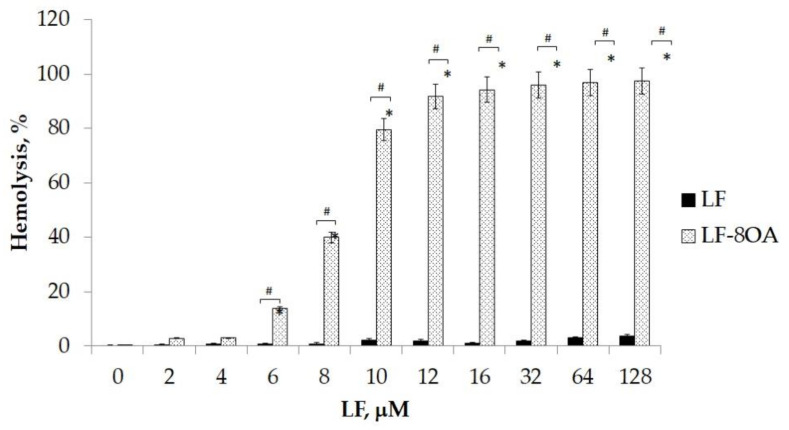
The hemolytic effect of LF/8OA compared with LF. The grade of hemolysis was assessed after 1 h of incubation of human erythrocytes with LF and LF/8OA at 37 °C. Control samples contained PBS or deionized water when complete lysis was needed. *—significant difference (*p* < 0.01) when compared with intact cells. #—significant difference (*p* < 0.01) when compared with LF without OA.

**Figure 4 ijms-24-16711-f004:**
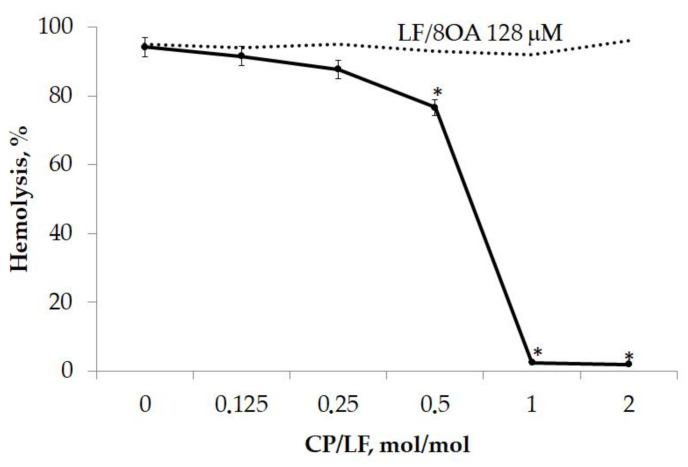
Effect of CP on the hemolytic activity of LF/8OA. The concentration of LF/8OA was 128 μM and remained the same during the entire experiment. The grade of hemolysis was assessed after 1 h of incubation of human erythrocytes with LF/8OA in the presence of various CP concentrations at 37 °C (line 1). *—significant difference (*p* < 0.05) compared with LF/8OA.

**Figure 5 ijms-24-16711-f005:**
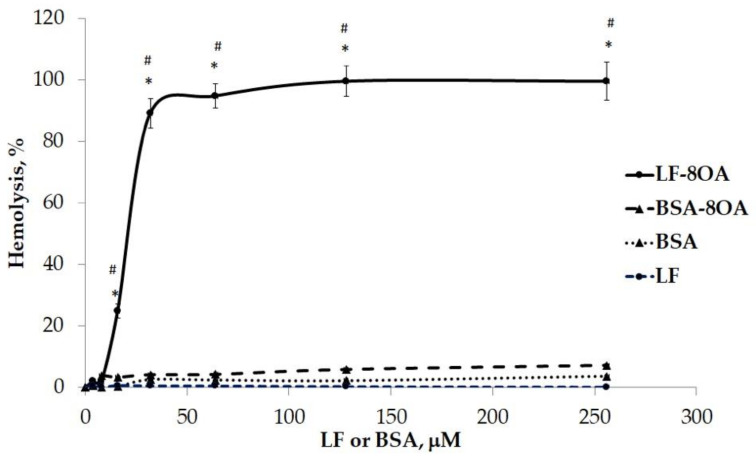
Comparison of the hemolytic effects of LF/8OA and BSA/8OA. The grade of hemolysis was assessed after 1 h incubation of human erythrocytes with LF (line 4), BSA (line 3), LF/8OA (line 1), and BSA/8OA (line 2) at 37 °C. Control samples contained PBS or deionized water when complete lysis was needed. *—significant difference (*p* < 0.01) when compared with the protein without OA. #—significant difference (*p* < 0.01) when compared with BSA/8OA.

**Figure 6 ijms-24-16711-f006:**
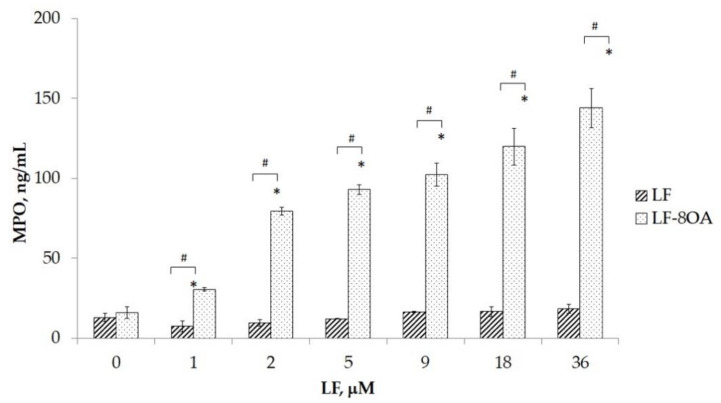
MPO concentration after 1 h incubation of human neutrophils with various concentrations of LF and LF/8OA. MPO in the supernatant was measured using ELISA with monoclonal antibodies to MPO (clone MPO18) and monoclonal antibodies labeled with horseradish peroxidase (clone 2F7). *—significant difference (*p* < 0.01) when compared with intact cells. #—significant difference (*p* < 0.01) when compared with LF without adding OA.

**Figure 7 ijms-24-16711-f007:**
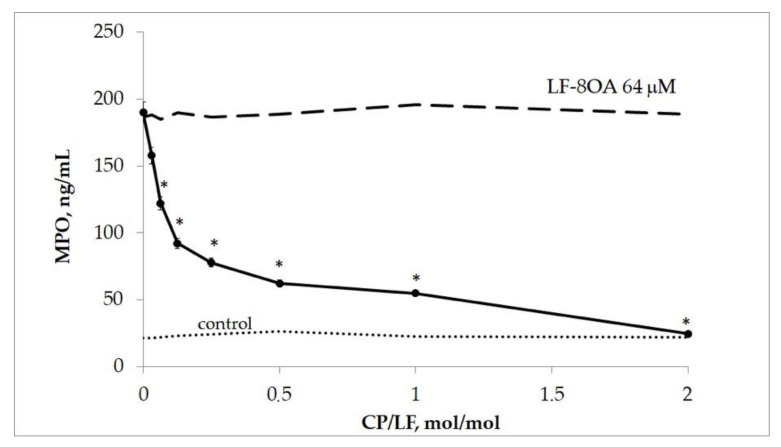
MPO concentration after 1 h of incubating human neutrophils with LF/8OA in the presence of various CP concentrations (line 1). The concentration of LF/8OA 64 μM remained the same during the entire experiment. *—significant difference (*p* < 0.05) as compared with LF/8OA.

**Figure 8 ijms-24-16711-f008:**
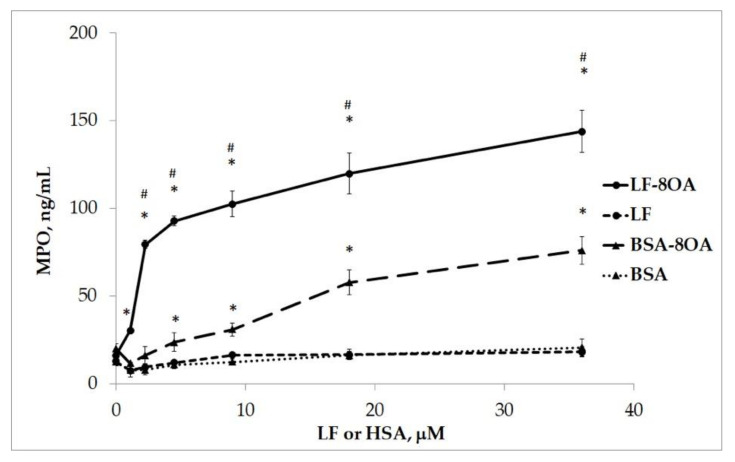
MPO concentration after 1 h of incubating human neutrophils with LF (line 4), LF/8OA (line 1), BSA (line 3), BSA/8OA (line2). *—significant difference (*p* < 0.01) when compared with control proteins (without adding OA). #—significant difference (*p* < 0.01) when compared with BSA/8OA.

**Table 1 ijms-24-16711-t001:** Parameters of affinity in interaction between LF (LF/8OA) and CP immobilized on CM5 chip. Data presented as X_m_ ± SE.

Analyte	Concentration, μM	k_a_, 10^4^/m × c	k_d_, 1/c	R_max_, RU	K_D_, μM	U-Value
LF	3.125–25.0	9.048 ± 1.1	0.03324 ± 0.0036	(23.74 ± 0.18) ×10^3^	0.367	7
LF/8OA		7.532 ± 0.58	0.03069 ± 0.002	(22.34 ± 0.12) ×10^3^	0.407	4

**Table 2 ijms-24-16711-t002:** Myeloperoxidase concentration upon the incubation of human neutrophils with LF/8OA and BSA/8OA. Data presented as X_m_ ± SE.

Protein Concentration, μM	LF/8OA	BSA/8OA
	MPO Concentration, ng/mL	
0	16.1 ± 3.5	20.3 ± 2.4
0.6	14.45 ± 1.0	15.9 ± 2.7
1.1	30.5 ± 2.4	13.6 ± 4.9
2.3	79.4 ± 2.9	16.3 ± 5.2
4.6	92.9 ± 7.2	23.4 ± 3.7
9.1	102.4 ± 11.5	31.1 ± 7.1
18.3	119.9 ± 12.1	57.9 ± 7.8
36.5	143.9 ± 9.5	76.1 ± 6.1

## Data Availability

The data presented in this study are available on request from the corresponding author.

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
