# Peer review of "Ceruloplasmin Reduces the Lactoferrin/Oleic Acid Antitumor Complex-Mediated Release of Heme-Containing Proteins from Blood Cells"

_ijms, 2023, doi:10.3390/ijms242316711_

Round 1

Reviewer 1 Report

Comments and Suggestions for Authors

The title of the article is misleading and does not reflect the information described.

The English must be revised in all document.

The results should be more robust and better explained.

The article is poorly written, presents a confusing English, specially in Materials and Methods, with lack of information necessary to understand the isolation and purification of ceruloplasmin (2.1).

Point 2.4 is very confusing.

In 2.5, the reading of A414nm is not explained!

Figure1- disproportionate and without molecular markers!

Figure 2- No interpretive color legend

References without standardization

For all these reasons the article is not in a position to be accepted.

Comments on the Quality of English Language

The English must have major revision in all document.

Author Response

RESPONSES TO REVIEWERS.

The authors are grateful to all reviewers for profound analysis and very useful suggestions. As was required, we modified the title and wrote a new Abstract. Please, see below our responses to the respected reviewers (point by point).

Reviewer 1.

Description of CP homogeneity testing has been added to the text.

Point 2.4 now is more clear.

Figure 1 now contains arrows showing positions of CP its complex with LF. Mr markers are inapplicable in case of non-denaturing electrophoresis, pictured in Fig. 1.

Figure 2. Color bars corresponding to the curves in the plot have been added.

References are brought to the same standard.

Reviewer 2.

  • Abstract has been rewritten and now seems to be more comprehensible.

2) The hypothesis is now clearly expressed at the end of Introduction.

Arguments in favor of the research accomplished with LF/8OA and ceruloplasmin as main targets are contained in the phrase: “Since mice receiving daily 5 mg of LF/8OA had no manifestations of that drug’s toxicity after 22 days, we decided to study the selectivity of the effect provided by that complex or to reveal a factor reducing its cytotoxicity”.

  • The Reviewer’s requirement was followed. Now information on reagents is incorporated in the text where necessary.

4) The English language has been improved by our colleague Robin McGuire, a native speaker.

5) We added a couple of phrases at the end of Discussion to make clear the overall implications of our findings. However, it is better no to separate Results and Discussion. It is much easier for a reader to get an explanation of the authors’ viewpoint on each result once the latter is mentioned, rather than look for such in a separate section.

6) Data on the approval by the local Ethics Committee are added in the text.

7) Data on immobilization have been added to the Methods.

8) The authors cannot provide any information concerning the use of the substance in patients, since no clinical study was done.

9) Please, see the responses to point 2 and 5.

Reviewer 3.

  • In our numerous preparatory experiments CP was thoroughly mixed with OA, but no interaction occurred.
  • A required experiment is not likely to be carried out lege artis, since OA is insoluble at concentrations used in the study. However, BSA/8OA has meager hemolytic effect on red blood cells, which is shown in Figure 5. This result speaks in favor of a very weak, if any, capacity of OA to cause hemolysis.
  • The same can be said about testing OA as possible activator of neutrophils (insoluble at high concentration). Similarly, OA in complex with BSA had much less pronounced effect, shown in Figure 8.

Reviewer 2 Report

Comments and Suggestions for Authors

Summary: In the present study, the authors assessed the toxicity of lactoferrin (LF) and LF complexed with oleic acid (LF/8OA) on human and mouse erythrocytes and neutrophils. They then determined that ceruloplasmin, which can interact with LF and LF/8OA, is a factor reducing cytotoxicity. The paper is difficult to follow and should be rewritten for clarity.  Specific comments are included below:

1)      The abstract will need to be rewritten. Please state the problem and provide background, present your findings, and state the overall significance of the research. Make your writing as clear and accessible as possible.

2)      The introduction could use more information. There is no clearly stated hypothesis, and it’s difficult to understand the rationale explaining why the authors performed the study.

3)      In materials and methods, the information in lines 62-87 should be incorporated into the proper subsection where the reagents were used.

4)      The manuscript could use an English language check.

5)      Please separate the results and discussion section and elaborate in the discussion about the overall implications of your findings.

6)      Was this study approved by your local IACUC?

7)      Please provide more details about the biosensors based on surface plasmon resonance in the methods section.

8)      Could the authors talk more about the clinical study data and how these compounds are being used in patients?

9)      More information needs to be provided to understand the rationale and significance of the study.

Comments on the Quality of English Language

Improper use of articles before nouns and improper plurality throughout the manuscript. 

Author Response

(The authors gave the same response as above.)

Reviewer 3 Report

Comments and Suggestions for Authors

The article written by A.Yu. Elizarova et al. presents that cytotoxic effect of the lactoferrin/oleic acid complex is reduced by ceruloplasmin. The manuscript can be accepted for publication after major revision. The authors should revise the manuscript according to the following comments.

1. (Figure 2) Interaction of ceruloplasmin with oleic acid should be investigated.

2. (Figure 3) Hemolytic effect of oleic acid on lysis of erythrocytes should be added.

3.  (Figure 6) Activation of neutrophils in the presence of oleic acid should be investigated.

Comments on the Quality of English Language

Minor editing of English language required.

Author Response

(The authors gave the same response as above.)

Round 2

Reviewer 1 Report

Comments and Suggestions for Authors

Some of the suggested recommendations were not carried out, such as:

- In Materials and Methods (electrophoretic methods), they describe an SDS-PAGE, but it doesn't exist in the doc.

- Fig. 2, is not treated correctly for a manuscript!

- References 24, 27 and 28 are still not standardized.

- In Fig.2 the graphs are not self-explanatory. The concentration of LF/8OA should be identified. The authors said they have added slashes to Fig. but they are not visible.

- In Fig.5, line 3 is not identified correctly.

- Authors should be eliminate the personnal pronoun "WE" in the abstract and introduction. One should not personalize the manuscript.

English should be further improved!

Comments on the Quality of English Language

English should be further improved!

Author Response

The authors are grateful to all reviewers for profound analysis and very useful suggestions. As was required, we modified the title and wrote a new Abstract. Please, see below our responses to the respected reviewers (point by point).

In response to the comments of Reviewer 1:

SDS-PAGE was used essentially to obtain evidence on homogeneity of CP and LF used in experiments. Hence, its results are not shown.

Fig. 2 was amended, and now the concentration of LF/8OA is identified.

Now in Fig. 5 all lines, including Line3, are designated.

The pronoun "we"has been eliminated from the abstract. Few times it is met in the entire text, but only when it is required to facilitate reading.

The authors made their best to brush up their English.

Reviewer 1.

Description of CP homogeneity testing has been added to the text.

Point 2.4 now is more clear.

Figure 1 now contains arrows showing positions of CP its complex with LF. Mr markers are inapplicable in case of non-denaturing electrophoresis, pictured in Fig. 1.

Figure 2. Color bars corresponding to the curves in the plot have been added.

References are brought to the same standard.

Reviewer 3 Report

Comments and Suggestions for Authors

Accept in present form.

Author Response

  • In our numerous preparatory experiments CP was thoroughly mixed with OA, but no interaction occurred.
  • A required experiment is not likely to be carried out lege artis, since OA is insoluble at concentrations used in the study. However, BSA/8OA has meager hemolytic effect on red blood cells, which is shown in Figure 5. This result speaks in favor of a very weak, if any, capacity of OA to cause hemolysis.
  • The same can be said about testing OA as possible activator of neutrophils (insoluble at high concentration). Similarly, OA in complex with BSA had much less pronounced effect, shown in Figure 8.
